# *TINF2* is a haploinsufficient tumor suppressor that limits telomere length

Isabelle Schmutz[1], Arjen R Mensenkamp[2], Kaori K Takai[1], Maaike Haadsma[2], Liesbeth Spruijt[2], Richarda M de Voer[2], Seunga Sara Choo[3], Franziska K Lorbeer[3], Emma J van Grinsven[3], Dirk Hockemeyer[3,4], Marjolijn CJ Jongmans[2†*], Titia de Lange[1*]

[1]Laboratory for Cell Biology and Genetics, Rockefeller University, New York, United States; [2]Department of Human Genetics, Radboud University Medical Center, Nijmegen, Netherlands; [3]Department of Molecular and Cellular Biology, University of California, Berkeley, Berkeley, United States; [4]Chan Zuckerberg Biohub, San Francisco, United States

*For correspondence:
M.C.J.Jongmans-3@umcutrecht.nl (MCJJ);
delange@mail.rockefeller.edu (TL)

Present address: † Princess Máxima Center for Pediatric Oncology, Department of Genetics, University Medical Center Utrecht, Utrecht, Netherlands

**Abstract** Telomere shortening is a presumed tumor suppressor pathway that imposes a proliferative barrier (the Hayflick limit) during tumorigenesis. This model predicts that excessively long somatic telomeres predispose to cancer. Here, we describe cancer-prone families with two unique *TINF2* mutations that truncate TIN2, a shelterin subunit that controls telomere length. Patient lymphocyte telomeres were unusually long. We show that the truncated TIN2 proteins do not localize to telomeres, suggesting that the mutations create loss-of-function alleles. Heterozygous knock-in of the mutations or deletion of one copy of *TINF2* resulted in excessive telomere elongation in clonal lines, indicating that *TINF2* is haploinsufficient for telomere length control. In contrast, telomere protection and genome stability were maintained in all heterozygous clones. The data establish that the *TINF2* truncations predispose to a tumor syndrome. We conclude that *TINF2* acts as a haploinsufficient tumor suppressor that limits telomere length to ensure a timely Hayflick limit.

## Introduction

The idea that telomere attrition could repress the outgrowth of early stage cancer originates from the observation that telomeres shorten in normal human cells (*Harley et al., 1990*; *Hastie et al., 1990*; *de Lange et al., 1990*; reviewed in *Maciejowski and de Lange, 2017*). In agreement with this theory, telomere shortening leads to a proliferative barrier in vitro (the Hayflick limit *Shay and Wright, 2000*) that can be overcome when telomerase is activated through expression of hTERT *Bodnar et al., 1998*; telomerase activity is required to create tumorigenic derivatives from normal human cells *Hahn et al., 1999*; and telomerase activation is a hallmark of human cancer (*Shay and Bacchetti, 1997*). The discovery of hTERT promoter mutations in familial melanoma and other tumor types further solidified the view that telomere attrition is a barrier to tumorigenesis (*Horn et al., 2013*; *Huang et al., 2013*; reviewed in *Lorbeer and Hockemeyer, 2020*).

For the telomere tumor suppression pathway to limit cancer incidence, telomeres need to shorten at the correct rate, which in most primary human cells is ~30–100 bp/end/cell division (*Harley et al., 1990*; *Huffman et al., 2000*). The number of cell divisions a transformed clone can execute before proliferation is curbed by one or more critically short telomeres depends on the initial telomere length. Most likely, it is the lengths of the shortest telomeres in a clone that determine its proliferative potential (*Hemann et al., 2001*; *Zou et al., 2004*). These considerations predict that excessive telomere length at birth will delay the Hayflick limit and create a permissive state for cancer development. Indeed, after removal of telomerase, a cancer cell line with very long telomeres remained

tumorigenic until its telomere reserve was depleted (*Taboski et al., 2012*). At birth, human telomeres have an average length that is specific to our species (*Kipling and Cooke, 1990*; *Gomes et al., 2011*). It is thought that this telomere reserve is sufficient to sustain the cell division needed for normal development and tissue homeostasis but becomes depleted during the over-proliferation associated with tumorigenesis. When and how human telomere length homeostasis is achieved has been difficult to discern.

In telomerase-positive tissue culture cells, telomere length homeostasis is mediated by shelterin (reviewed in *Hockemeyer and Collins, 2015*). TIN2 is a central component in shelterin that binds to three other shelterin subunits. TIN2 interacts with both double-stranded telomeric DNA-binding proteins – TRF1 and TRF2 – and binds TPP1, which forms a heterodimer with the single-stranded telomeric DNA-binding protein POT1 (reviewed in *de Lange, 2018*). TIN2 has been implicated as negative regulator of telomere length as have TRF1 and POT1 (*van Steensel and de Lange, 1997*; *Kim et al., 1999*; *Loayza and De Lange, 2003*). The current model for telomere length homeostasis invokes a negative feedback loop, wherein telomerase is inhibited in cis by proteins (e.g. TRF1, TIN2, and POT1) that accumulate on the TTAGGG repeats synthesized by the enzyme.

It is well established that when telomeres are too short at birth, a devastating bone-marrow failure syndrome (dyskeratosis congenita [DC] and related syndromes) can arise. Missense mutations in a short stretch of amino acids of TIN2 (the DC patch) are responsible for ~25% of DC cases (*Savage et al., 2008*; *Walne et al., 2008*; reviewed in *Savage and Bertuch, 2010*), whereas the majority of DC cases are due to mutations impinging on telomerase biogenesis and activity.

Recent data on cancer-associated mutations in *POT1* have provided a hint that long telomeres may predispose to cancer. Inherited *POT1* mutations in cancer-prone families are associated with excessively long telomeres in somatic cells (*Robles-Espinoza et al., 2014*; *NCI DCEG Cancer Sequencing Working Group et al., 2014*; reviewed in *Gong et al., 2020*). However, the *POT1* mutations also lead to genome instability, which has been invoked as the main pathogenic determinant (*Ramsay et al., 2013*; *Pinzaru et al., 2016*; *Chen et al., 2017*; *Gu et al., 2017*). Therefore, the *POT1* mutations have not provided unambiguous evidence for the idea that long telomeres predispose to cancer.

Here, we describe heterozygous loss-of-function mutations in *TINF2* in cancer-prone families. These mutations do not compromise telomere protection but create excessively long telomeres in vitro and in vivo. We conclude that the affected individuals are cancer-prone because their overly long telomeres thwart the telomere tumor suppressor pathway.

## Results

### Germline *TINF2* mutations in families with cancer

In a routine diagnostic setting, whole-exome sequencing was performed on lymphocyte DNA of patients who developed multiple malignancies and/or had a striking family history of cancer. Germline variants in exon 5 of *TINF2* (encoding TIN2) were discovered in four probands (*Figure 1A–C*; *Figure 1—figure supplement 1*). Three probands shared c.604G > C, whereas the fourth carried c.557del. The six individuals in this study developed 14 malignancies (*Figure 1A*), including three papillary thyroid carcinomas, three breast carcinomas, and two melanomas (*Figure 1A*). No loss of heterozygosity was detected in six tumors tested and second hits in *TINF2* were excluded in four of the six tumors analyzed by whole-exome sequencing (F3:III-1; Astrocytoma, F2:II-1; Melanoma and breast cancer, F1:II-4; colorectal cancer (CRC), see also *Figure 1—figure supplement 2*). Multiple somatic driver mutations were identified, all previously associated with the tumor type in which the mutation was identified, such as *BRAF* (c.1799T > A, p.Val600Glu) in CRC and melanoma, and *PIK3CA* (c.1624G > A, p.Glu542Lys) in breast cancer (*Figure 1—figure supplement 2*). The tumors did not reveal a shared somatic mutational spectrum (data not shown). Based on these families, we suggest that carriers of the reported *TINF2* variants might benefit from regular thyroid and dermatological surveillance as well as more general cancer surveillance.

Both *TINF2* mutations generated truncated proteins (*Figure 2*). The c.557del mutation creates a shift in the reading frame after serine 186 that ends in a stop codon 23 amino acids downstream (p. (Ser186fs); *Figure 2D*). The c.604G > C change disrupts the splice donor site of exon 5 (*Figure 2A, B*). Transcript analysis showed that in addition to the wild-type full-length transcript

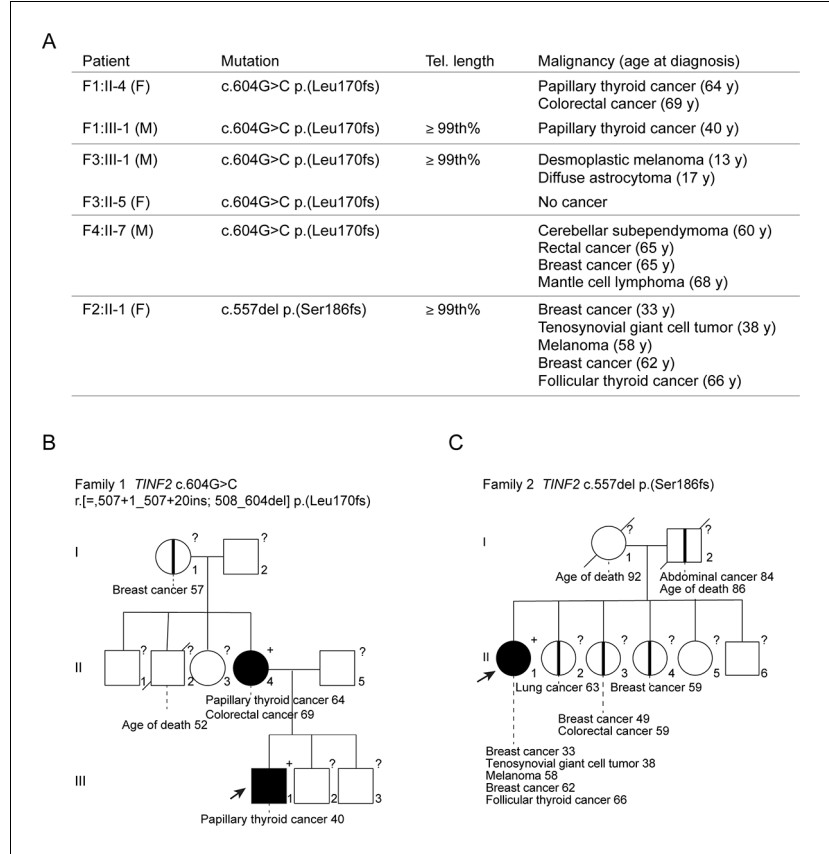

**Figure 1.** Germline mutations in *TINF2* identified in individuals with multiple malignancies. (**A**) *TINF2* mutations and clinical features of affected individuals in four different families. Telomere length percentile is based on Flow-FISH data (see below *Figure 5—figure supplement 1A*). (**B, C**) Pedigrees of one of the c.604G > C families (**B**) and the c.557del family (**C**) listed in (**A**). Probands are highlighted by arrows. Filled symbols indicate patients with confirmed *TINF2* mutations and their clinical features are indicated. Symbols with vertical lines denote individuals who have developed cancer but have not been tested for *TINF2* mutations. +: *TINF2* mutation; -: wild type for *TINF2*; ?: not tested. See also *Figure 1—figure supplement 1*.

The online version of this article includes the following figure supplement(s) for figure 1:

**Figure supplement 1.** Pedigrees of two c.604G > C *TINF2* families.

**Figure supplement 2.** Somatic mutations in the COSMIC cancer gene census identified in malignancies in *TINF2* mutation carriers.

(ENST00000399423.8) and a transcript lacking exons 4 and 5 (ENST00000626689.2), patient samples contain an alternative transcript (604G > C II, *Figure 2A,B*). Transcript c.604G > C II lacks exon 5 and contains 20 extra nucleotides from intron 4. It appears to arise from an alternative splice donor site in intron four with a good splice site consensus score (alt D4; *Figure 2B,C*). This transcript was also observed in heterozygous RPE1 cells carrying the c.604G > C change (see below). In addition, the +/c.604G > C RPE1 cells contained a second allele-specific transcript (c.604G > C I, *Figure 2B*) generated through an alternative donor site in intron five that bears a good splice site consensus sequence (alt D5; *Figure 2B,C* and see *Figure 4—figure supplement 1* below). The use of alt D5 results in the addition of 17 nucleotides from intron 5 (*Figure 2B*). This transcript was most likely missed in the analysis of the patient samples due to its lower abundance and co-migration with the wild-type full-length transcript. The c.604G > C I and c.604G > C II transcripts both have a frameshift in the TIN2 ORF and are predicted to encode truncated proteins (p.(L170fs) and p.(E202fs)) (*Figure 2D*).

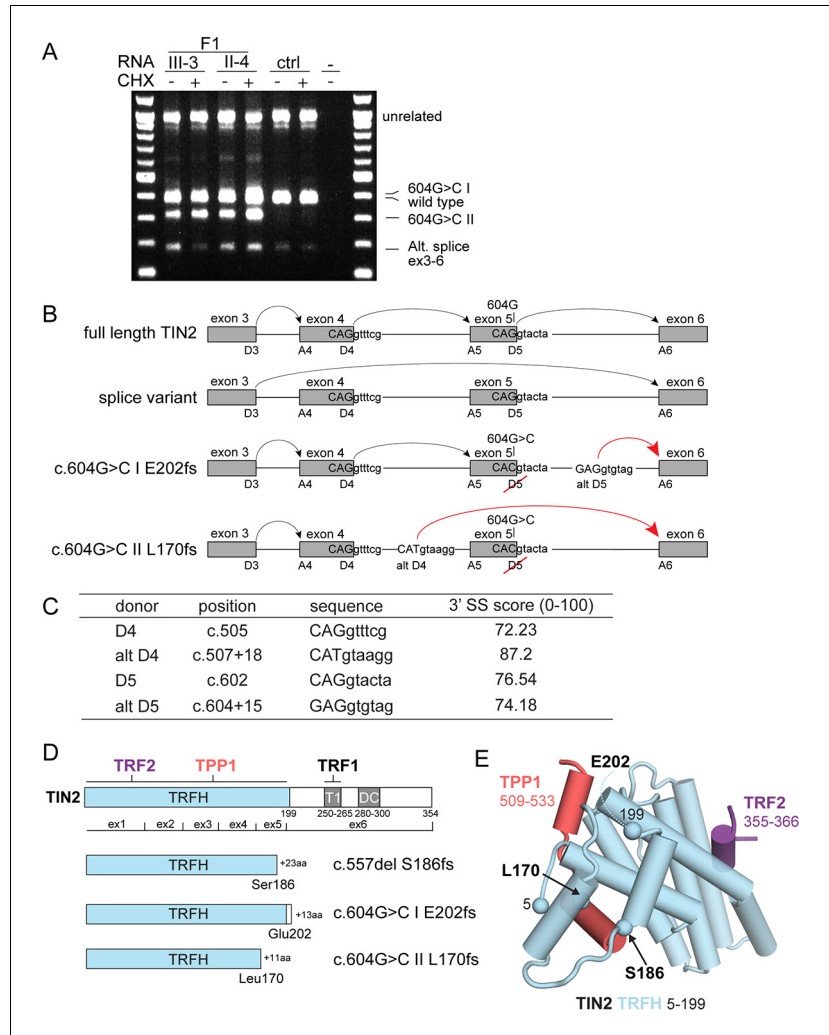

**Figure 2.** Molecular analysis of transcripts resulting from *TINF2* mutations. (**A**) Transcript analysis in peripheral blood lymphocytes (with and without cycloheximide treatment, CHX) from patients with the c.604G > C *TINF2* mutation (F1:III-3 and F1:II-4; see *Figure 1A*) and a control individual. RT-PCR products were analyzed by Sanger sequencing. Wild-type full-length TIN2 mRNA, an alternative splice form found in wild-type cells (alt. splice exons 3–6) and mutant allele transcripts (604G > C I and 604G > C II) are indicated. Transcript 604G > C I was identified in heterozygous +/c.604G > C and homozygous c.604G > C RPE1 cells. (**B**) Schematic showing the splicing of exons 3–6 for full-length wild-type *TINF2*, the alternative splice variant (exons 3–6), and the aberrant splicing occurring in cells with c.604G > C mutations. Alt D4 and alt D5 indicate alternative splice donor sites. (**C**) Comparison of the consensus score of alternative splice donor sites alt D4 and alt D5 to splice donors D4 and D5 (as calculated by Human Splicing Finder www.umd.be). (**D**) Schematic of wild-type TIN2, and the predicted truncations resulting from expression of c.557del p.(S186fs), c.604G > C I p.(E202fs), and c.604G > C II p.(L170fs). Exon boundaries and the regions involved in TIN2 interactions with TRF1, TRF2, and TPP1 and the DC patch are indicated. (**E**) Structure of the TIN2 TRFH domain (PDB ID: 5xyf; *Hu et al., 2017*) with the amino acids at the truncation points highlighted. Peptides from TPP1 and TRF2 that interact with the TRFH domain are shown in the structure.

## Truncated versions of TIN2 do not bind TRF1 and do not localize to telomeres

The predicted truncated TIN2 proteins contain most of the N-terminal TRF homology (TRFH) domain of TIN2 where TRF2 and TPP1 bind and lack the TRF1-binding site and the short patch of amino acids mutated in dyskeratosis congenita (DC patch *Savage et al., 2008*; *Walne et al., 2008*; *Figure 2D,E*). To determine whether the truncated TIN2 proteins retain interactions with

TIN2's binding partners in shelterin, we generated expression constructs for the three predicted TIN2 truncations: L170fs and E202fs from c.604G > C and S186fs from c.557del. Co-immunoprecipitation from 293T cells co-transfected with HA-tagged TIN2 versions and myc-tagged TRF1 showed that, as expected, the truncated forms of TIN2 had lost the ability to bind to TRF1 (*Figure 3A,B*). The interaction with TRF2 was preserved in the c.604G > C derived E202fs truncation and was apparently enhanced in the c.557del-derived S186fs truncation (*Figure 3A,B*). In contrast, the c.604G > C L170fs protein showed very little (or no) interaction with TRF2 (*Figure 3A,B*). The

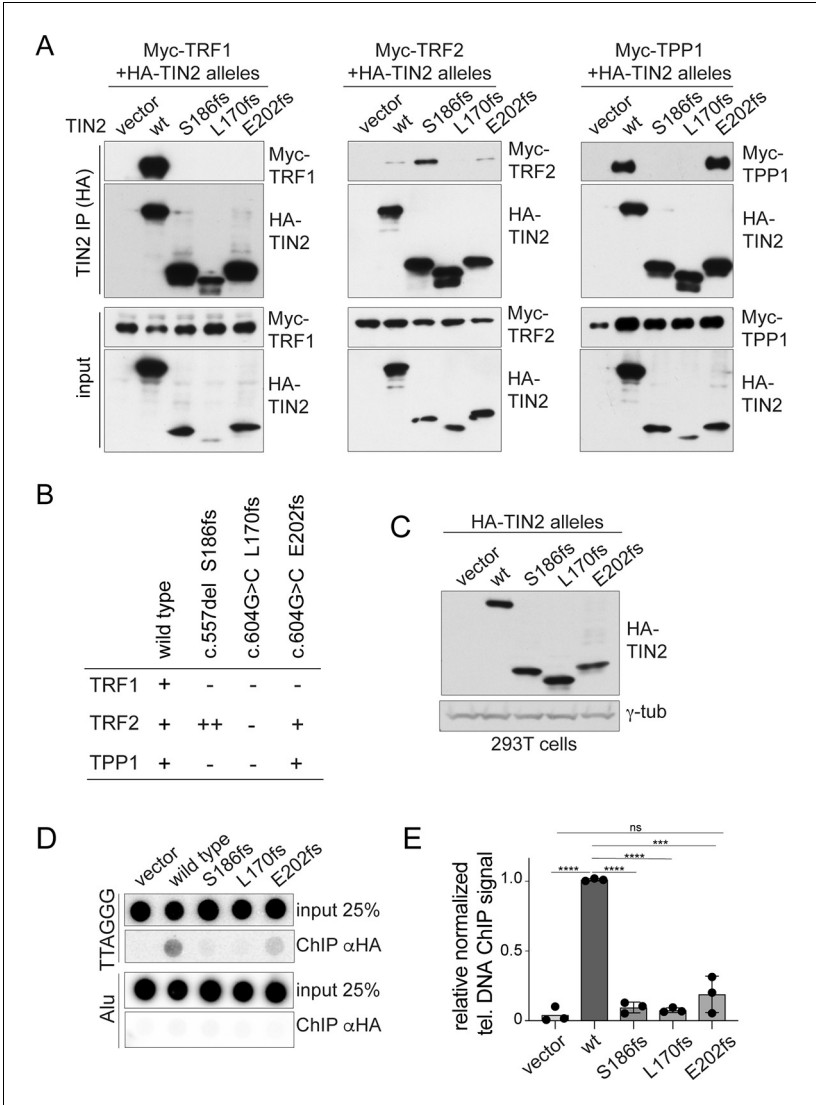

**Figure 3.** Truncated TIN2 versions show altered binding to shelterin subunits and diminished telomeric localization. (**A**) Co-immunoprecipitation of myc-tagged TRF1 (left panel), TRF2 (middle panel) and TPP1 (right panel) from 293T cells co-transfected with HA-tagged wt TIN2, S186fs, L170fs, E202fs, or the empty vector. Inputs and HA-IPs were probed with HA antibody to detect TIN2 and with myc antibody to detect TRF1, TRF2, and TPP1. To achieve equal expression levels, the ratio of plasmids was: wt 1x, 186fs 2.5x, 202fs 2.5x, and 170fs 5x. This experiment was repeated three times with comparable results. (**B**) Summary of the interaction of wild type and mutant TIN2 alleles with TRF1, TRF2, or TPP1 as derived from multiple co-IP experiments as in (**A**). (**C**) Immunoblot showing expression of HA-tagged wild type and mutant TIN2 versions in 293T cells used for telomeric ChIP. (**D**) Dot blot assay for telomeric ChIP performed on the indicated 293T cells as shown in (**C**). (**E**) Quantification of telomeric DNA recovered with HA Ab (average relative % telomeric DNA recovered in three independent experiments, individual data points and means ± SD are shown). For the quantification, unpaired t-test was used to determine significance, p-values: ****p<0.0001, ***p<0.001, **p<0.01, *p<0.05. ns, not significant.

interaction with TPP1 was preserved in the E202fs version of TIN2 but not in the two other truncated forms (*Figure 3A,B*).

Since the localization of TIN2 to telomeres is primarily determined by its ability to interact with TRF1 (*Frescas and de Lange, 2014*), it is expected that the truncated proteins fail to efficiently accumulate at telomeres. In agreement, telomeric ChIP assays using the HA antibody on chromatin from 293T cells expressing the HA-tagged versions of TIN2 showed that the S186fs and L170fs proteins do not associate with telomeric DNA (*Figure 3C–E*). The E202fs protein may be slightly more proficient in telomeric association but the fraction of telomeric DNA recovered in the ChIP was not significantly increased compared to cells transfected with the empty vector. These data indicate that the truncated versions of TIN2 have lost the ability to function at telomeres.

## Telomeres are fully protected in cells heterozygous for c.604G > C or c.557del

CRISPR/Cas9-mediated gene editing was used to knock in the c.557del and c.604G > C mutations in RPE1-hTERT cells deficient for Rb and p53 (*Yang et al., 2017*). The mutations were introduced using knock-in repair ssODN templates with the desired mutation, a mutated PAM, and a restriction enzyme recognition site used for screening of clonal cell lines (*Figure 4—figure supplement 2*). The CRISPR/Cas9 editing was designed to generate matched wild-type control clones with *TINF2* genes that were either unedited or edited with silent nucleotide changes (introduction of a restriction enzyme site and a mutated PAM). In addition, we targeted exon 1 of *TINF2* to generate heterozygous KO clones (TIN2+/KO or +/-; *Figure 4—figure supplement 3*) and accompanying control clones. For each genotype, several clones with the desired alterations were isolated (*Figure 4—figure supplement 4*). The mutated clonal cell lines showed the same proliferation rate as the control clones (*Figure 4—figure supplement 5A*). CRISPR/Cas9 editing of RPE1 cells also yielded a viable clone homozygous for the c.604G > C mutation (*Figure 4—figure supplement 4B*). Although this clone does not represent the genotype of the patients, it is useful as a positive control in telomere dysfunction assays. Transcript analysis of the heterozygous and homozygous c.604G > C RPE1 clones confirmed expression of the c.604G > C II mRNA identified in patient samples (see *Figure 2A,B*) and identified the c.604G > C I transcript as an additional product from the mutated locus (*Figure 4—figure supplement 1* and *Figure 2B*).

Clones heterozygous for the c.604G > C or c.557del mutation and the TIN2+/- clones had slightly lower TIN2 protein levels relative to the controls (*Figure 4A,B*). Based on telomeric chromatin immunoprecipitation (ChIP) and immunofluorescence (IF) analysis, cells heterozygous for the mutations retained TIN2, TRF1, TPP1, and POT1 at their telomeres, although the data do not exclude a moderate reduction in the association of these proteins with telomeres (*Figure 4—figure supplement 5B–G*). In contrast, the homozygous c.604G > C clone showed a complete absence of TIN2 in immunoblots and the presence of TIN2, TRF1, TPP1, and POT1 at telomeres was strongly reduced (*Figure 4A,B* and *Figure 4—figure supplement 5E–G*).

The extent to which the *TINF2* mutations affected telomere protection was monitored via the telomere dysfunction induced foci (TIF) assay (*Takai et al., 2003*), which measures the accumulation of 53BP1 at telomeres. Clones with heterozygous c.557del or c.604G > C mutations had the same TIF response as the control cells (*Figure 4C,D* and *Figure 4—figure supplement 6A*). Similarly, the TIF response was not increased in the TIN2+/- clones compared to wild-type controls. In contrast, the homozygous c.604G > C clone showed obvious loss of telomere protection (*Figure 4C,D* and *Figure 4—figure supplement 6A*), likely due to the reduced telomeric association of POT1, which is required to prevent the activation of ATR signaling at human telomeres (*Denchi and de Lange, 2007*). These results indicate that one functional *TINF2* gene is sufficient to sustain full telomere protection and that the truncated TIN2 proteins do not have a dominant negative effect on telomere protection.

Similarly, analysis of metaphase spreads showed that heterozygosity for the *TINF2* mutations or the exon 1 KO allele did not induce a significant level of telomere dysfunction. Although cells carrying the homozygous c.604G > C mutation showed elevated levels of sister telomere associations and a low level of chromosome fusions, such aberrations were not significantly increased in the heterozygous clones relative to wild type (*Figure 4E–G*, *Figure 4—figure supplement 6B,C*). Thus, *TINF2* is not haploinsufficient for telomere protection and the *TINF2* mutations are unlikely to induce cancer-promoting genome rearrangements.

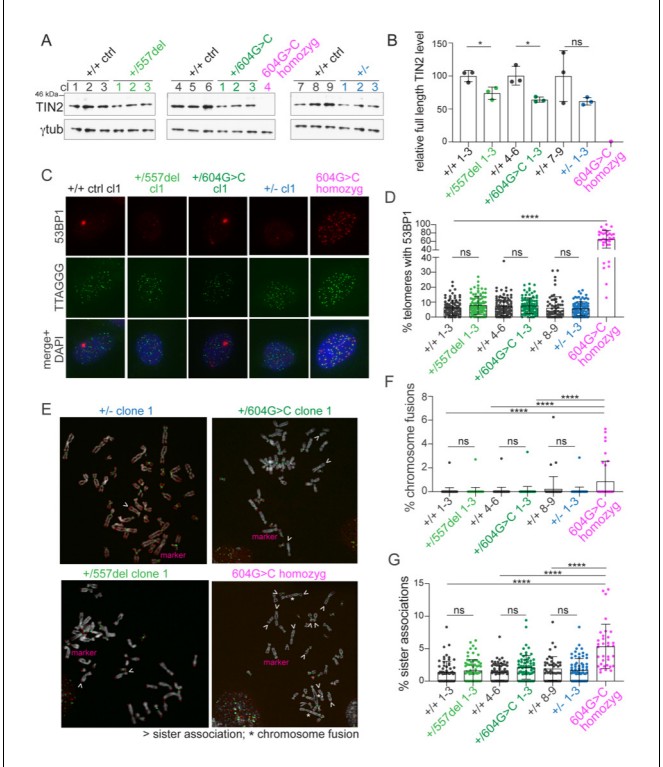

**Figure 4.** Heterozygous *TINF2* mutations do not cause telomere damage or genome instability. (**A**) Immunoblot for TIN2 and γtubulin in control cells and the indicated clones with targeted *TINF2* alleles. (**B**) Quantification of the immunoblot shown in A. Unpaired t-test was used to determine significance. Symbols: *p<0.05; ns, not significant (0.16). (**C**) Representative images of TIF analysis in control and indicated *TINF2* mutant cells. IF for 53BP1 (red), telomeric FISH (green) and DNA (DAPI, blue). (**D**) Quantification of percentage of telomeres colocalizing with 53BP1 foci. Data from ≥50 nuclei per cell line, with three cell lines per genotype (with the exception of the single c.604G > C homozyg clone). (**E**) Representative metaphase spreads of cells with mutated *TINF2* alleles. Sister telomere associations (>), telomere fusions (*), and a marker chromosome found in all clones (marker) are indicated. Telomere FISH (red), centromere FISH (green) and DNA (DAPI, gray). (**F**) Quantification of telomere fusions ≥20 spreads per cell line, with three cell lines per genotype (except for the single 604G > C homozyg clone). (**G**) Quantification of the % of telomeres found in sister associations. Data from ≥20 spreads per cell line; three cell lines per condition, except for the single 604G > C homozyg clone. For the quantification in (**B**), (**D**), (**F**), and (**G**) means ± SD and individual data points are shown. One-way ANOVA with Tukey post-test was used to determine significance, p-values: ****p<0.0001, ***p<0.001, **p<0.01, *p<0.05. ns, not significant. See also *Figure 4—figure supplements 1–6*.

The online version of this article includes the following figure supplement(s) for figure 4:

**Figure supplement 1.** Transcript analysis in 604G > C/+ cells reveals presence of two alternative *TINF2* transcripts (604G > C I, 604G > C II).

**Figure supplement 2.** Knock-in strategy for introduction of c.557del and c.604G > C mutations into RPE1 cells.

**Figure supplement 3.** Strategy to generate TIN2+/- RPE1 clones.

**Figure supplement 4.** Sanger sequencing of CRISPR/Cas9-engineered clones with *TINF2* mutations.

**Figure supplement 5.** Characterization of cells with targeted *TINF2* alleles.

**Figure supplement 6.** Representation of TIFs, telomere fusions, and sister associations in the individual cell lines.

## Excessive telomere elongation associated with c.604G > C and c.557del

Telomere length analysis in lymphocytes from three patients carrying the c.604G > C or c.557del mutations revealed a median telomere length above the 99th percentile as measured by Flow-FISH (*Figure 1A* and *Figure 5—figure supplement 1A*). Similarly, individuals with the *TINF2* p.W198fs mutation showed telomeres that were approximately twofold longer based on qPCR (*He et al., 2020*). The presence of exceptionally long telomeres in the c.604G > C individuals was verified by

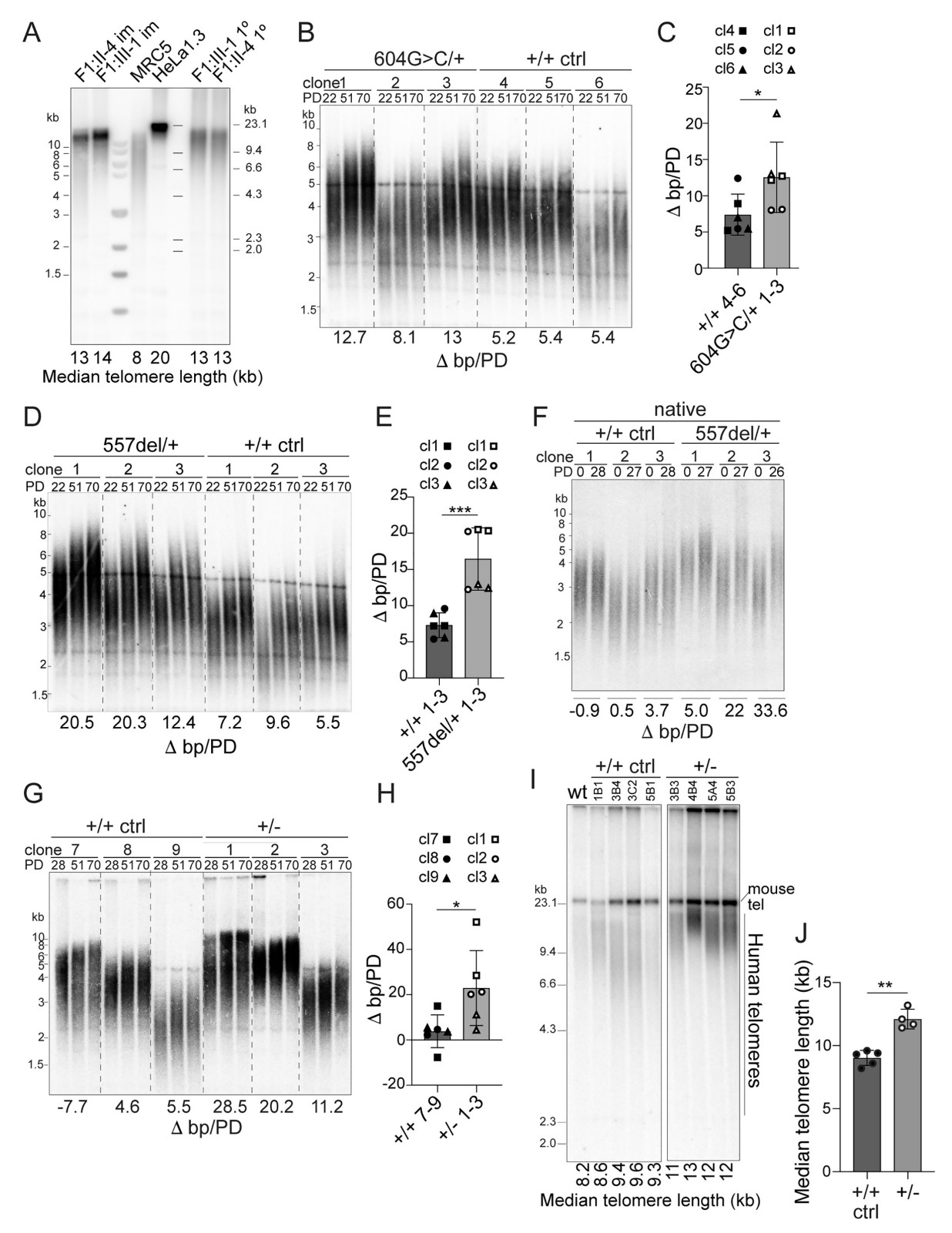

**Figure 5.** Heterozygous *TINF2* mutations induce telomere lengthening. (**A**) Telomeric Southern blot of *MboI/AluI*-digested genomic DNA from immortalized and primary patient cells (lymphocytes), normal lung fibroblasts (MRC5), and HeLa1.3 cells. Median telomere length (MTL) is indicated. (**B**) Telomeric Southern blot of *MboI/AluI*-digested genomic DNA from control clones and clones with heterozygous c.604G > C mutations at the indicated PDs. Telomere length changes are indicated and were calculated over 48 PDs. (**C**) Quantification of median telomere length changes for control cells

*Figure 5 continued on next page*

*Figure 5 continued*

and cells with heterozygous c.604G > C mutations. Three cell lines per genotype were analyzed in two independent experiments (symbols denote the individual cell lines). (D) Telomeric Southern blot as in (B) for control clones and clones with heterozygous c.557del mutations. (E) Quantification of median telomere length changes for control cells and cells with heterozygous c.557del mutations as in (C). (F) Detection of telomeres in *MboI/AluI*-digested genomic DNA from control clones and clones with heterozygous c.557del mutation probed under native conditions with a telomeric probe for the 3′ overhang. The change in MTL over 28 PDs is indicated. (G) Telomeric southern blot as in (B) for control cells and TIN2+/- cells. The indicated telomere length changes were calculated over 42 PDs. (H) Quantification of median telomere length changes for control cells and TIN2+/- cells as in C. (I) Telomeric southern blot of *MboI/AluI*-digested genomic DNA from control and TIN2+/- hESCs. All clones were generated and propagated in parallel and telomere length was determined at 28 days after the CRISPR/Cas9 targeting. (J) Quantification of the median telomere length (as determined by blotting as in (I)) for control and heterozygous hESCs clones (control, n = 5; TIN2+/KO, n = 4). Bar graphs in (C), (E), (H), and (J) show means ± SDs. *P*-values are based on unpaired t-test. ****p<0.0001, ***p<0.001, **p<0.01, *p<0.05. ns, not significant. See also *Figure 5—figure supplements 1–3*.

The online version of this article includes the following figure supplement(s) for figure 5:

**Figure supplement 1.** Long telomeres in *TINF2* c.604G > C and c.557del patients and hESC CRISPR/Cas9 editing.

**Figure supplement 2.** No evidence for increased telomere recombination in c.604G > C mutant and Tin2+/- cells.

**Figure supplement 3.** Mutant and control RPE1 clones show similar telomerase activity.

genomic blotting, showing that both primary and EBV-immortalized lymphocytes from two patients carried telomeres of ~13 kb (*Figure 5A*).

Consistent with the unusually long telomeres in the patients, telomere elongation was observed in RPE1 clones heterozygous for c.557del or c.604G > C (*Figure 5B–F*). For each genotype, three cell lines were tested over up to 3 months of propagation with the appropriate control cell lines cultured and analyzed in parallel. The change in telomere length with population doublings (PDs) was measured in two independent experiments for each clone. As expected, the initial telomere lengths show clonal variation as has been observed in other cell lines (*Bryan et al., 1998*; *Takai et al., 2010*). Comparison of the telomere elongation per PD between the control clones and the heterozygous clones showed that both the c.557del and the c.604G > C mutation resulted in a greater extension of the telomeres (*Figure 5B–F*). This was the case when the total telomeric DNA was detected in standard genomic blots and the same result was obtained when telomere length was evaluated based on hybridization of a probe to the 3′ overhang in native gels (*Figure 5F*). Similarly, cells heterozygous for the exon one truncation showed greater rates of telomere elongation and this phenotype was observed in RPE1 cells as well as in human embryonic stem cells (hESCs) heterozygous for a deletion of exons 4–7 (*Figure 5G–J* and *Figure 5—figure supplement 1B,C*). All RPE1 clones showed approximately the same telomerase activity (*Figure 5—figure supplement 2*). The observed telomere elongation is likely due to telomerase-mediated elongation because there was no evidence for increased telomere recombination in the cell lines (*Figure 5—figure supplement 3*), and the telomeres did not show the typical heterogeneous size of ALT telomeres (*Figure 5*). An effect of TIN2 on telomerase-mediated telomere elongation is consistent with prior data showing that a truncated TIN2 protein induced dramatic telomere elongation in cells expressing telomerase but had no effect in telomerase-negative cells (*Kim et al., 1999*). These data indicate that the *TINF2* gene is haploinsufficient for telomere length control and explain the telomere elongation phenotype in the patients.

## Discussion

These data reveal *TINF2* as a haploinsufficient tumor suppressor gene. Inactivation of one *TINF2* allele through truncation mutations results in inherited cancer predisposition with high penetrance and severity. The spectrum of cancers in the *TINF2* families studied here is broad, including breast cancer, colorectal cancer, thyroid cancer, and melanoma, and several patients had multiple independent malignancies. Similarly, a recent report identified a *TINF2* truncation mutation in a large family affected by thyroid cancer and melanoma (*He et al., 2020*). It is remarkable that inherited mutations in *TINF2* can have two widely distinct outcomes. While the loss-of-function mutations described here cause cancer through aberrant telomere elongation, missense mutations in the DC patch of *TINF2* cause bone-marrow failure syndromes that are due to excessive telomere shortening. These disparate outcomes reflect the dual role of TIN2, which uses its DC patch to promote telomere

maintenance by telomerase, while keeping telomere length in check through its interaction with TRF1 and TPP1/POT1. Our genetic data underscore that telomere length at birth needs to be carefully controlled within a narrow range to prevent premature stem cell depletion on one hand and cancer on the other.

## Genetic evidence for the telomere tumor suppressor pathway

It has proven difficult to test the idea that telomere shortening represents a tumor suppressor pathway. Apart from modeling in the mouse (*Artandi and DePinho, 2000*), evidence in favor of this decades-old concept is derived largely from indirect or in vitro observations (reviewed in *Maciejowski and de Lange, 2017*). The discovery of hTERT promoter mutations in familial melanoma and sporadic cancers argued that telomerase activation is a critical step in tumor progression (*Horn et al., 2013*; *Huang et al., 2013*; reviewed in *Lorbeer and Hockemeyer, 2020*). However, it did not definitively establish that telomerase is needed to subvert the telomere tumor suppression pathway. The requirement for telomerase activation during cancer progression could also be due to its presumed ability to heal broken chromosomes arising during periods of genome instability.

Cancer-predisposing mutations in the POT1 subunit of shelterin also did not inform on the telomere tumor suppressor pathway because they have two outcomes of potential relevance to cancer. On one hand, the altered *POT1* alleles result in very long germline telomeres in the probands, consistent with the idea that telomere shortening curbs tumorigenesis (*NCI DCEG Cancer Sequencing Working Group et al., 2014*; *Robles-Espinoza et al., 2014*). On the other hand, the mutations were reported to induce genome instability (*Ramsay et al., 2013*; *Pinzaru et al., 2016*; *Chen et al., 2017*; *Gu et al., 2017*). Indeed, most reports concluded that the cancer predisposition associated with these POT1 alleles is due to genomic rearrangements. However, these studies largely relied on overexpression of mutant versions of POT1 and did not examine cells with the heterozygous POT1 mutations found in the patients.

The cancer-causing *TINF2* mutations that create long germline telomeres without affecting telomere protection now remove the ambiguity. In the *TINF2* cases affected by inherited cancer predisposition, it is extremely unlikely that genome instability contributes to tumorigenesis since we have not detected loss of telomere protection or genome instability in heterozygous cell lines whose genotype mimic the patient status. The simplest interpretation is that the patient's frequent malignancies are due to a failure in the telomere tumor suppressor pathway. By extension, we argue that there is no need to invoke genome instability as a cancer-promoting aspect of the *POT1* mutations in familial cancer. The data presented here argue that the telomere elongation phenotype associated with the *POT1* mutations is sufficient to explain the higher incidence of cancer in these families.

According to our findings, exceptionally long telomeres can lead to cancer predisposition. Our conclusion is consistent with prior GWAS studies suggesting an effect of telomere length on cancer predisposition (*Rode et al., 2016*; *Telomeres Mendelian Randomization Collaboration et al., 2017*; reviewed in *McNally et al., 2019*). It is therefore pertinent to consider measuring telomere length in cancer-prone families lacking other genetic risk factors. Our findings also suggest that caution is warranted with regard to efforts to interfere with the telomere shortening program in healthy individuals (*Harley et al., 2011*).

## Telomere length homeostasis in vitro and in vivo

The elongated telomeres associated with the inherited *POT1* and *TINF2* mutations now suggest that the telomere length homeostasis observed in cultured cells reflect aspects of telomere length control in the human germline. In vitro, TRF1, TIN2, and POT1 have been implicated as negative regulators of telomere length largely based on the telomere elongation phenotype of dominant negative alleles (*van Steensel and de Lange, 1997*; *Kim et al., 1999*; *Loayza and De Lange, 2003*). The role of TPP1 (which links POT1 to TIN2) is more complex because it acts to limit telomere extension through POT1 but also recruits telomerase to telomeres (*Nandakumar et al., 2012*; *Zhong et al., 2012*). How the shelterin subunits control telomerase activity in cis has remained opaque (reviewed in *Hockemeyer and Collins, 2015*). It is clear that the trimeric Ctc1, Stn1, Ten1 (CST) complex is required to control the length of human telomeres but how CST blocks telomerase is still unknown (*Wan et al., 2009*; *Chen et al., 2012*; *Feng et al., 2017*; *Takai et al., 2016*). The finding that cell culture systems reflect regulatory pathways observed in vivo should spur further in vitro experiments

designed to illuminate how telomere length homeostasis is achieved. An important question to be addressed is why TIN2 is haploinsufficient for telomere length control and which other shelterin components show this phenotype. Shelterin components that are haploinsufficient for telomere length control but not for telomere protection are of particular interest since loss-of-function mutations in these genes could predispose to cancer.

### When does TIN2 act as a tumor suppressor?

*TINF2* mutations lead to unusually long telomeres in the peripheral blood lymphocytes of the patients reported here. It is assumed that such long telomeres reflect the long telomeres present in the bone marrow stem cells since the number of divisions separating bone marrow stem cells from peripheral lymphocytes is too limited to allow the low level of telomerase in lymphocytes to extend the telomeres substantially. If this assumption is correct, the patients are likely born with unusually long telomeres in most of their stem cell compartments.

How could such long stem cell telomeres have originated? One possibility is that the *TINF2* mutations lead to extended telomeres in the parental germline that are then inherited by the affected child. We argue that this scenario does not account for several observations. First, if long telomeres inherited from one parent were the cause of the cancer predisposition, all children of an affected parent would be equally predisposed to cancer. Our data and the co-segregation of the *TINF2* p. W198fs variant with cancer in one large family (*He et al., 2020*), argues that this is not the case. Second, genomic blots indicate a single class of very long telomeres in the peripheral blood lymphocytes of adults with the *TINF2* mutations. If these individuals had inherited long telomeres from one parent and short telomeres from the other without further changes in telomere length (except for the usual telomere attrition), the telomeres should reveal two size classes, one of which is in the normal range. Our genomic blots show that this is not the case. Finally, simple inheritance of long telomeres from one parent would not delay the onset of the Hayflick limit, which is determined by the shortest telomeres in a clone. The normal-sized telomeres from the unaffected parent should allow the Hayflick limit to protect against cancer regardless of the presence of longer telomeres from the affected parent.

These considerations lead us to propose that the *TINF2* mutations act by inappropriately elongating telomeres during early development. Importantly, such a process would elongate the normal sized telomeres inherited from the unaffected parent, preventing the timely onset of the Hayflick limit. Since telomerase is detectable in a number of embryonic tissues during the first and second trimester (*Wright et al., 1996*), it is reasonable to assume that tens of cell divisions take place during which the enzyme can elongate telomeres unless it is restrained by the telomere length homeostasis pathway. We imagine that the *TINF2* mutations exert their cancer-promoting effects in the first weeks or months after fertilization, resulting not only in long telomeres in the germline but also in all other stem cell compartments that are relevant to cancer development later in life. According to this reasoning, *TINF2* would be a tumor suppressor gene with a very specific window of opportunity, exerting its effect early in development but not later.

## Materials and methods

**Key resources table**

| Reagent type (species) or resource | Designation | Source or reference | Identifiers | Additional information |
| --- | --- | --- | --- | --- |
| Cell line (*H. sapiens*) | 293T | ATCC | | |
| Cell line (*H. sapiens*) | hTERT-RPE1 p53-/- Rb-/- | *Yang et al., 2017* | | |
| Cell line (*H. sapiens*) | hTERT-RPE1 p53-/- Rb-/- +/604G > C.1 clone 1–3 m | This paper | | Heterozygous for *TINF2* c.604G > C |
| Cell line (*H. sapiens*) | hTERT-RPE1 p53-/- Rb-/- +/604G > C.2 clone 2–23 m | This paper | | Heterozygous for *TINF2* c.604G > C |
| Cell line (*H. sapiens*) | hTERT-RPE1 p53-/- Rb-/- +/604G > C.3 clone 1–21 m | This paper | | Heterozygous for *TINF2* c.604G > C |

*Continued on next page*

Continued

| Reagent type (species) or resource | Designation | Source or reference | Identifiers | Additional information |
|---|---|---|---|---|
| Cell line (*H. sapiens*) | hTERT-RPE1 p53-/- Rb-/- ctrl4 clone 1–4 c | This paper | | Control cell line |
| Cell line (*H. sapiens*) | hTERT-RPE1 p53-/- Rb-/- ctrl5 clone 1–13 c | This paper | | Control cell line |
| Cell line (*H. sapiens*) | hTERT-RPE1 p53-/- Rb-/- ctrl6 clone 5–8 c | This paper | | Control cell line |
| Cell line (*H. sapiens*) | hTERT-RPE1 p53-/- Rb-/- +/557del.1 clone 1–9 m | This paper | | Heterozygous for *TINF2* c.557del |
| Cell line (*H. sapiens*) | hTERT-RPE1 p53-/- Rb-/- +/557del.2 clone 1–14 m | This paper | | Heterozygous for *TINF2* c.557del |
| Cell line (*H. sapiens*) | hTERT-RPE1 p53-/- Rb-/- +/557del.3 clone 2–17 m | This paper | | Heterozygous for *TINF2* c.557del |
| Cell line (*H. sapiens*) | hTERT-RPE1 p53-/- Rb-/- ctrl1 clone 2–7 c | This paper | | Control cell line |
| Cell line (*H. sapiens*) | hTERT-RPE1 p53-/- Rb-/- ctrl2 clone 2–8 c | This paper | | Control cell line |
| Cell line (*H. sapiens*) | hTERT-RPE1 p53-/- Rb-/- ctrl3 clone 1–2 c | This paper | | Control cell line |
| Cell line (*H. sapiens*) | hTERT-RPE1 p53-/- Rb-/- +/-.1 clone 1-1het | This paper | | Heterozygous for *TINF2* |
| Cell line (*H. sapiens*) | hTERT-RPE1 p53-/- Rb-/- +/-.2 clone 1-3het | This paper | | Heterozygous for *TINF2* |
| Cell line (*H. sapiens*) | hTERT-RPE1 p53-/- Rb-/- +/-.3 clone 2-4het | This paper | | Heterozygous for *TINF2* |
| Cell line (*H. sapiens*) | hTERT-RPE1 p53-/- Rb-/- +/+ ctrl7 clone 2–9 c | This paper | | Control cell line |
| Cell line (*H. sapiens*) | hTERT-RPE1 p53-/- Rb-/- +/+ ctrl8 clone 3–2 c | This paper | | Control cell line |
| Cell line (*H. sapiens*) | hTERT-RPE1 p53-/- Rb-/- +/+ ctrl9 clone 1–4 c | This paper | | Control cell line |
| Cell line (*H. sapiens*) | WIBR3 hESC | *Lengner et al., 2010* | NIH stem cell registry number: 0079 | Wild-type |
| Cell line (*H. sapiens*) | WIBR3 hESC 1B1 | This paper | | +/+ *TINF2* Control cell line |
| Cell line (*H. sapiens*) | WIBR3 hESC 3B4 | This paper | | +/+ *TINF2* Control cell line |
| Cell line (*H. sapiens*) | WIBR3 hESC 3C2 | This paper | | +/+ *TINF2* Control cell line |
| Cell line (*H. sapiens*) | WIBR3 hESC 5B1 | This paper | | +/+ *TINF2* Control cell line |
| Cell line (*H. sapiens*) | WIBR3 hESC 3B3 | This paper | | +/- *TINF2* |
| Cell line (*H. sapiens*) | WIBR3 hESC 4B4 | This paper | | +/- *TINF2* |
| Cell line (*H. sapiens*) | WIBR3 hESC 5A4 | This paper | | +/- *TINF2* |
| Cell line (*H. sapiens*) | WIBR3 hESC 5B3 | This paper | | +/- *TINF2* |
| Anti-hTIN2 | | *Ye and de Lange, 2004* | #864 | |
| Anti-γTubulin | | Sigma | GTU88 | |
| Anti-Myc | | Cell signaling | 9B11 | |

*Continued*

| Reagent type (species) or resource | Designation | Source or reference | Identifiers | Additional information |
|---|---|---|---|---|
| Anti-HA | | Abcam | Ab9110 | |
| Anti-53BP1 | | Abcam | ab175933 | |
| Cy3-OO- (TTAGGG)$_3$ | | PNA bio | | Telomere probe |
| FITC-OO-(CCCTAA)$_3$ | | PNA bio | | Telomere probe |
| Alexa Fluor 647-OO-(TTAGGG)$_3$ | | PNA bio | | Telomere probe |
| CENPB-AF488 | | PNA bio | F3004 | Centromere probe |

No statistical methods were used to predetermine sample size. Key resources are listed in the key resource table.

## Patient selection

As part of routine diagnostic procedure, whole exome sequencing and subsequent cancer predisposition panel analysis was performed on patients who developed multiple malignancies or had a striking family history of cancer. The four index patients reported in this study were sequenced between 2014 and 2019 and were part of a total cohort of 446 patients referred for this diagnostic procedure. Whole exome sequencing was performed with relevant clinical quality accreditations and consent procedures as approved by the IRB equivalent (Medisch Etische Toetsingscommissie) of the Radboud University Medical Center. All participants (four probands and two affected relatives) provided written informed consent for publication of their data.

## Whole exome sequencing

Genomic DNA was isolated from whole blood. The experimental workflow of all exomes was performed at BGI Europe (Beijing Genome Institute Europe, Copenhagen, Denmark). Exonic regions were enriched using the Agilent (Agilent Technologies, CA, USA) SureSelect V4 (*n* = 85) or V5 (*n* = 169) kit and sequenced using an Illumina Hiseq (Illumina, CA, USA) sequencer with 101 bp paired end reads to a median coverage of >75 x. Sequenced reads were mapped to the hg19 reference genome using the mapping algorithm from BWA (*Li and Durbin, 2010*) (version 0.5.9-r16) and called by the GATK unified genotyper (*McKenna et al., 2010*) (version 3.2–2). All variants were annotated using an in-house pipeline for exome analysis containing variant and gene-specific information. This information includes the variant population frequencies from >5000 in-house whole exome analyses performed (*Lelieveld et al., 2016*).

Whole genome sequencing was performed on tumor DNA as described previously (*NTHL1 study group et al., 2020*) using the SureSelectXT Human All Exon V6 enrichment kit (Agilent Technologies, CA, USA) on a NextSeq500 sequencing platform (Illumina, CA, USA). Trimmed NextSeq 500 sequencing reads were aligned to hg19 by using BWA-MEM, and duplicates were flagged by using Picard Tools, version 1.90. Variants were called with Mutect2 (GATK version 4.1.0.0), with matched germline samples; variant filtering was performed as described (*NTHL1 study group et al., 2020*). All variants were annotated using an in-house annotation pipeline and driver genes were selected based in the COSMIC cancer gene census.

## Exome variant interpretation

For the gene panel analysis, a bioinformatic in silico filter was applied to select for variants affecting the known cancer predisposition genes. This gene panel consisted of 114 established (OMIM) cancer predisposition genes in 2013, expanding to 232 genes in 2019. [https://www.radboudumc.nl/getmedia/59c91c86-e6e0-433b-995a-4e91b8277572/HEREDITARY-CANCER-PANEL_DG217.aspx]. All subsequent versions of this panel included the *TINF2* gene, because of its role in cancer predisposition in dyskeratosis congenita. Variants were filtered for coding, non-synonymous variants with population frequencies below 1% in our in-house database and evaluated regarding their possible pathogenicity. The latter was performed using population frequencies, nucleotide conservation scores (PhyloP), and in silico pathogenicity predictions (SIFT, Polyphen2, Mutation Taster).

## Transcript analysis

For transcript analysis in c.604G > C mutant and control cells, RNA isolation (RNeasy Mini kit, Qiagen) and cDNA synthesis (Superscript IV Reverse Transcriptase, ThermoFisher) was performed according to standard protocols. *TINF2* transcripts were amplified, separated according to size, cloned into pCR-Topo and analyzed by Sanger Sequencing. Primers used:

    TINF2_transcriptfw (TINF2-exon2) 5'- TCCTGAAAGCCCTGAATCAC-3'
    TINF2_transcriptrv (TINF2-exon6) 5'-GGGTCTGGCATGGACTCTTA-3'.

## Cell culture

293T cells (ATCC; not further authenticated) were grown in DMEM supplemented with 10% bovine calf serum (Hyclone), 2 mM L-glutamine, 100 U/ml penicillin, 0.1 mg/ml streptomycin, and 0.1 mM nonessential amino acids. hTERT-RPE1 p53-/- Rb-/-cells (*Yang et al., 2017*) were generated using RPE1 cells from the ATCC (not further authenticated) and cultured in DMEM/F12 (Gibco) supplemented with 10% fetal bovine serum (GIBCO), 100 U/ml penicillin (Sigma) and 0.1 mg/ml streptomycin (Sigma). Stem cell culture was performed as described previously (*Chiba et al., 2015*) using WIBR#3 hESCs (NIH stem cell registry number: 0079; not further authenticated *Lengner et al., 2010*). All cell lines were free of mycoplasma.

## Expression vectors

Vectors expressing N-terminally Flag-(HA)$_2$ tagged TIN2 and N-terminally Myc-tagged TRF1, TRF2, and TPP1 were as previously described (*Smogorzewska and de Lange, 2002*; *Frescas and de Lange, 2014*). Flag-(HA)$_2$-TIN2 S186fs was cloned by site-directed mutagenesis (QuikChange II XL, Agilent Technologies). For the cloning of Flag-(HA)$_2$-TIN2 L170fs and Flag-(HA)2-TIN2 E202fs, mutant transcripts were amplified from the cDNA recovered from c.604G > C mutant cells (604G > C homozyg) and the wild-type fragment in pLPC-Flag-(HA)$_2$-Tin2 was replaced with the respective mutant *BamHI/BlpI* fragments.

## Co-immunoprecipitation and immunoblotting

For co-immunoprecipitation assays, Flag-(HA)$_2$ tagged TIN2 proteins were co-expressed with Myc-tagged TRF1, TRF2 or TPP1 in 293T cells. Cells were collected 36–48 hr after calcium phosphate transfection as previously described (*Takai et al., 2016*). Lysates were diluted to lower the NaCl concentration to 200 mM for TPP1 and TRF1 and to 100 mM for TRF2 immunoprecipitations. HA-tagged TIN2 was precipitated using αHA agarose beads for 2 hr at 4°C, beads were washed with lysis buffer and PBS, proteins were eluted with Laemmli loading buffer and analyzed by immunoblotting using aHA antibody (HA.11, Covance) for TIN2 and Myc antibody (9B11, Cell signaling) for co-immunoprecipitated TRF1, TRF2, and TPP1. For TIN2 immunoblots, whole-cell lysates were prepared by lysis of cells in buffer C (20 mM Hepes-KOH pH 7.9, 0.42 M KCl, 25% glycerol, 0.1 mM EDTA, 5 mM MgCl$_2$, 0.2% NP-40, complete protease inhibitor cocktail), quantified by Biuret protein assay and immunoblotted using antibodies for human TIN2 (#864) and γtubulin (GTU88, Sigma).

## Telomeric ChIP

Telomeric ChIP was performed as previously described (*Loayza and De Lange, 2003*). Telomeric DNA associated with shelterin proteins was immunoprecipitated with the following crude sera or purified antibodies: crude rabbit TRF1 (#371), crude rabbit TIN2 (#865), crude rabbit TPP1 (#1151), POT1 (Abcam, ab123784), anti-HA (Abcam, ab9110) and protein G magnetic beads (Cell signaling). For ChIP of exogenously introduced TIN2 alleles, 293T cells were transfected by calcium phosphate transfection, and crosslinked and harvested 36–48 hr after transfection.

## CRISPR/Cas9-mediated targeting of *TINF2* in RPE-1 cells

Clonal cell lines with targeted *TINF2* alleles were generated using pU6-(BbsI)-Cbh-Cas9-T2a-mCherry (*Chu et al., 2015*) that allows co-expression of sgRNA and Cas9 linked to mCherry via the T2A peptide. For the knock-in of *TINF2* mutations, the Cas9-sgRNA expression vector (TINF2exon5, sgRNA-1 or sgRNA-2) was delivered together with a 1:1 mix of the appropriate donor oligonucleotides (ssODN) by electroporation (Lonza). mCherry-positive cells were selected by single-cell sorting.

Clones were screened by restriction enzyme digestion of PCR products and editing was verified by Sanger sequencing of Topo-cloned PCR products. For the generation of TINF2+/- cells, Cas9-sgRNA (TINF2exon1, sgRNA-3) was introduced by electroporation, mCherry-positive targeted cells were selected by single cell sorting. Clones were screened by Sanger sequencing of PCR products for introduction of mono-allelic indels creating frame-shift mutations in exon 1.

sgRNA oligonucleotides were purchased from ThermoFisher and cloned into BbsI-digested expression vector. The sequences are: sgRNA-1 5'-TTGTCTCCAGGCAAGAGAAG-(PAM)−3'; sgRNA-2 5'-GACAATATGGTGTGGACATG-(PAM)−3'; sgRNA-3 5'-ACGCCTTTGTATGGGCCTAA-(PAM)−3' ssODN were purchased from IDT and had the following sequences: c.557del-mut 5'-GCTTCAGGATGTGCTGAGTTGGATGCAGCCTGGAGTCTCTATCACTTCTTTCTTGCCTGGAGACAATATGGTGTAGACATGGGGTGGCTGCTTCCAGGTACTAGGAATTTGGAGGTGTAGTGTTTAGC-3'; c.557del-control 5'-GCTTCAGGATGTGCTGAGTTGGATGCAGCCTGGAGTCTCTATCACTTCTTCTCTTGCCTGGAGACAATATGGTGTTGACATGGGGTGGCTGCTTCCAGGTACTAGGAATTTGGAGGTGTAGTGTTTAGC-3'; c.604G > C mut 5'-CAGCTTCAGGATGTGCTGAGTTGGATGCAGCCTGGAGTCTCTATCACCTCTTCTCTTGCCTGGAGACAATATGGTGTAGACATGGGGATGGCTGCTTCCACGTACTAGGAATTTGGAGGTGTAGTGTTTAGCCTGAGACCTTTTGAGGCAGTCCACTGGAATAGTT-3'.

c.604G > C control 5'-CAGCTTCAGGATGTGCTGAGTTGGATGCAGCCTGGAGTCTCTATCACCTCTTCTCTTGCCTGGAGACAATATGGTGTTGACATGGGATGGCTGCTTCCAGGTACTAGGAATTTGGAGGTGTAGTGTTTAGCCTGAGACCTTTTGAGGCAGTCCACTGGAATAGTT-3'.

For screening, the following primers were used: PCRscreen_exon5fw 5'-GGCCACTAACCCACTTTTG-3'; PCRscreen_exon5rv 5'-CCTAGAGGGGCCAGATTGA-3'; PCRscreen_exon1fw 5'-TTCCGCGAGTACTGGAGTTT-3'; PCRscreen_exon1rv 5'-TCCCCTTCCAGGTCCTACTT-3'.

## CRISPR/Cas9-mediated targeting of *TINF2* in hESCs

Stem cell culture and editing experiments were performed as described previously (*Chiba et al., 2015*). To delete exons 4–7 of *TINF2* in WIBR#3 hESCs (NIH stem cell registry number: 0079; *Lengner et al., 2010*), cells were co-electroporated with 15 ug of two PX330 Cas9 plasmids (*Cong et al., 2013*), containing guide sequences 5'-TGTTCAAGTTCCTACAGCAG-3' and 5'-CCTGACTCAGACTACCTACC-3', respectively, and 7.5 µg of a GFP plasmid. Targeting was confirmed by PCR on genomic DNA using fw primer 5'-GGCCACTAACCCACTTTTGA-3' and rev primer 5'-TGGCCATTTTCTTCCTCATC-3' (Phusion, annealing temperature 63.4C, 1:15 min extension). Expected product sizes are 1275 bp for wild-type band and 218 bp for the exons 4–7 deletion.

## IF-FISH

For immunofluorescence in combination with telomeric FISH (IF-FISH), cells grown on coverslips to sub-confluence and were fixed in MeOH for 10 min at −20℃. IF-FISH was carried out as previously described (*Takai et al., 2003*). The following affinity purified antibodies were used for IF: rabbit TRF2 (#647), rabbit TRF1 (#371), rabbit TIN2 2 (#864), rabbit 53BP1 (Abcam ab175933). Telomeric DNA was detected with FITC-OO-(CCCTAA)$_3$ PNA probe. Images were captured on a DeltaVision microscope (Applied Precision) equipped with a cooled charge-coupled device camera (DV Elite CMOS Camera), a PlanApo 60 × 1.42 NA objective (Olympus America), and SoftWoRx software.

## FISH and CO-FISH on metaphase chromosomes

Telomeric FISH and CO-FISH were conducted as previously described (*van Steensel et al., 1998*; *Celli et al., 2006*) using Alexa Fluor 647-OO-(TTAGGG)3, Cy3-OO-(TTAGGG)three or FITC-OO-(CCCTAA)three and a centromere probe (PNA Bio). Images were captured using a DeltaVision microscope (Applied Precision) equipped with a cooled charge-coupled device camera (DV Elite CMOS Camera) and a PlanApo 60 × 1.42 NA objective (Olympus America), and controlled by and SoftWoRx software.

## Flow-FISH

Flow-FISH analysis was performed by RepeatDX (Aachen, Germany) on DNA from patient peripheral blood lymphocytes according to standard protocols (*Alter et al., 2007*).

## Telomere length analysis

For analysis of telomere length, cells were grown for 70 PDs and samples were harvested periodically by trypsinization, washed with 1x PBS, pelleted and frozen until further analysis. Genomic DNA was prepared as previously described (*de Lange et al., 1990*). DNA for telomere length analysis was digested with *Mbo*I and *Alu*I, quantified using Hoechst 33259 fluorometry and 0.5–1 μg was run on 0.7% agarose gels in 0.5x TBE. The DNA was depurinated, denatured, and neutralized and transferred onto membrane as previously described (*de Lange et al., 1990*). Blots were probed for telomeres using the Sty11 probe (*de Lange, 1992*). Alternatively, telomere length was evaluated based on the hybridization of a probe to the 3′ overhang in native gels. For this, gels were dried and probed with an end-labeled $(CCCTAA)_4$ as previously described (*Karlseder et al., 2002*). Gels and membranes were exposed to Phophorimager screens and quantified with Fiji.

## TRAP assay

TRAP assay was performed according to manufacturer's descriptions (TRAPeze Telomerase Detection Kit, EMD Millipore). Reaction products were run on a native polyacrylamide gel and stained with ethidium bromide.

## Acknowledgements

We thank the members of the de Lange lab for helpful discussion. John Zinder is thanked for generating the structural representation of the TIN2 truncations. Research reported in this publication was supported by grants from the NCI (R35CA210036), the Breast Cancer Research Foundation, and the Melanoma Research Alliance (MRA 577521) to T.d.L. T.d.L. is an American Cancer Society Rose Zarucki Trust Research Professor. D.H. is a Chan Zuckerburg Biohub Investigator, a Pew-Stewart Scholar for Cancer Research supported by the Pew Charitable Trusts and the Alexander and Margaret Stewart Trust. This research is supported by grants to D.H. from the Siebel Stem Cell Institute, the NIH (R01-CA196884), the D.O.D. (W81XWH-19-1-0586), and a Research Scholar Grants from the American Cancer Society (133396-RSG-19-029-01-DMC).

## Additional information

### Competing interests

Titia de Lange: Member of the SAB of Calico Life Sciences LLC. The other authors declare that no competing interests exist.

### Funding

| Funder | Grant reference number | Author |
| --- | --- | --- |
| NIH | R01-CA196884 | Dirk Hockemeyer |
| U.S. Department of Defense | W81XWH-19-1-0586 | Dirk Hockemeyer |
| American Cancer Society | 133396-RSG-19-029-01-DMC | Dirk Hockemeyer |
| National Cancer Institute | R35CA210036 | Titia de Lange |
| Melanoma Research Alliance | 577521 | Titia de Lange |
| American Cancer Society | Rose Zarucki Trust Research Professor | Titia de Lange |
| Breast Cancer Research Foundation | | Titia de Lange |
| Siebel Stem Cell Institute | | Dirk Hockemeyer |
| Pew Charitable Trusts | | Dirk Hockemeyer |
| Alexander and Margaret Stewart Trust | | Dirk Hockemeyer |

The funders had no role in study design, data collection and interpretation, or the decision to submit the work for publication.

### Author contributions

Isabelle Schmutz, Conceptualization, Data curation, Supervision, Validation, Investigation, Visualization, Methodology, Writing - original draft, Writing - review and editing; Arjen R Mensenkamp, Conceptualization, Data curation, Supervision, Investigation, Methodology, Writing - review and editing; Kaori K Takai, Data curation, Investigation; Maaike Haadsma, Liesbeth Spruijt, Data curation; Richarda M de Voer, Data curation, Investigation, Writing - review and editing; Seunga Sara Choo, Emma J van Grinsven, Investigation; Franziska K Lorbeer, Investigation, Writing - review and editing; Dirk Hockemeyer, Conceptualization, Data curation, Supervision, Funding acquisition, Investigation, Writing - original draft, Writing - review and editing; Marjolijn CJ Jongmans, Conceptualization, Supervision, Investigation, Writing - review and editing; Titia de Lange, Conceptualization, Data curation, Formal analysis, Supervision, Funding acquisition, Visualization, Methodology, Writing - original draft, Project administration, Writing - review and editing

### Author ORCIDs

Isabelle Schmutz https://orcid.org/0000-0001-7193-3922
Arjen R Mensenkamp https://orcid.org/0000-0003-3805-877X
Franziska K Lorbeer http://orcid.org/0000-0002-3152-6852
Dirk Hockemeyer http://orcid.org/0000-0002-5598-5092
Titia de Lange http://orcid.org/0000-0002-9267-367X

### Ethics

Human subjects: This study was performed in accordance with the local Institutional Review Board and all participants (four probands and two affected relatives) provided written informed consent for publication of their data.

### Decision letter and Author response

Decision letter https://doi.org/10.7554/eLife.61235.sa1
Author response https://doi.org/10.7554/eLife.61235.sa2

## Additional files

### Supplementary files

• Supplementary file 1. Uncropped images of immunoblots and telomere blots shown in the main figures.

• Supplementary file 2. Uncropped images of immunoblots and telomere blots shown in the main figures.

• Supplementary file 3. Uncropped images of immunoblots and telomere blots shown in the main figures.

• Supplementary file 4. p-Values and summary statistics for key data.

• Transparent reporting form

### Data availability

All data generated or analysed during this study are included in the manuscript and supporting files. Source data files have been provided in the Supplementary files.

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
