## [Decision Letter]

**Acceptance summary:**

The experiments presented in this manuscript directly probe and validate the previously assumed idea that a germline telomere over-elongation down the line permits more cell divisions to occur. These extra cell divisions will allow for more chances for other mutations to accumulate such that full-fledged cancer incidence increases and tumors arise quite early in life. Therefore, these results provide strong evidence that *TINF2* indeed is a haploinsufficient tumor suppressor gene that acts only through telomere length restriction. The ramifications of these findings go as far as strongly caution against the use of any manipulations aimed at elongating telomeres in people.

**Decision letter after peer review:**

Thank you for submitting your article "*TINF2* is a haploinsufficient tumor suppressor that limits telomere length" for consideration by *eLife*. Your article has been reviewed by three peer reviewers, one of whom is a member of our Board of Reviewing Editors, and the evaluation has been overseen by Richard White as the Senior Editor. The following individual involved in review of your submission has agreed to reveal their identity: Joachim Lingner (Reviewer #2).

The reviewers have discussed the reviews with one another and the Reviewing Editor has drafted this decision to help you prepare a revised submission.

I am happy to write that reviewers commented very positive about it. They did indeed feel that an important question was addressed and that the results were clear and telling. They agree with you in that they thought that conceptually; this work is very important as is it supports the notion that overly long telomeres promote tumor formation presumably by delaying the onset of cellular senescence.

However, there were also some suggestions and questions relating to the data. We are aware that much of your data are derived from patient samples and perhaps not available for further experimentation and that will of course be taken in consideration for the final decision.

1) If cancer sequencing is available, one might ask whether the cancers in the affected people have anything else in common? Undoubtedly additional mutations are required for malignant transformation. If there are no unusual commonalities, this should be stated, since it is a bold discovery that simple telomere elongation leads to malignancy predisposition.

2) Perhaps another prediction would be that in patients with *TINF2*-mutations premalignant lesions containing senescent cells should be unusually large. Or did the patients even carry tumors without upregulating hTERT expression?

3) Effects on telomere elongation by homologous recombination or the telomere shortening rate are not excluded. Therefore, it appears reasonable to address this issue. In order to test the roles of telomerase, the telomere length kinetics could be assessed upon deletion of a telomerase subunit or the inhibition of telomerase with the BIBR or GRN inhibitors.

---

## [Author Response]

[…] There were also some suggestions and questions relating to the data. We are aware that much of your data are derived from patient samples and perhaps not available for further experimentation and that will of course be taken in consideration for the final decision.1) If cancer sequencing is available, one might ask whether the cancers in the affected people have anything else in common? Undoubtedly additional mutations are required for malignant transformation. If there are no unusual commonalities, this should be stated, since it is a bold discovery that simple telomere elongation leads to malignancy predisposition.

We have now added exome sequencing data that indicates no special pattern of oncogenic mutations. The accompanying text is:

"No loss of heterozygosity was detected in six tumors tested and second hits in *TINF2* were excluded in four of the six tumors analyzed by whole-exome sequencing (F3:III-1; Astrocytoma, F2:II-1; Melanoma and breast cancer, F1:II-4; colorectal cancer (CRC), see also Figure 1—figure supplement 2). […] The tumors did not reveal a shared somatic mutational spectrum (data not shown)."

2) Perhaps another prediction would be that in patients with TINF2-mutations premalignant lesions containing senescent cells should be unusually large. Or did the patients even carry tumors without upregulating hTERT expression?

This is an interesting question but not one we can answer with the limited number of patient samples available. Hopefully, more insight into these issues will be obtained when more patients become available in the future.

3) Effects on telomere elongation by homologous recombination or the telomere shortening rate are not excluded. Therefore, it appears reasonable to address this issue. In order to test the roles of telomerase, the telomere length kinetics could be assessed upon deletion of a telomerase subunit or the inhibition of telomerase with the BIBR or GRN inhibitors.

To address the dependence of the telomere elongation phenotype on telomerase would take us a very long time. Because BIBR often only partially inhibits telomerase, we would have to do bulk CRISPR/Cas9 KO of hTERT or TERC in our clones (18 total). This will likely take months to optimize before we can initiate the telomere elongation experiments which themselves take several months. We note that TIN2 has been documented as a major regulator of telomerase-mediated telomere elongation by the Campisi lab (Kim et al., 1999) and others, making it likely that it acts through the telomerase pathway in our experiments.

However, we were able to address the alternative possibility that telomere elongation in our heterozygous cell lines is due to telomere recombination. We show that the level of T-SCEs (readout for telomere recombination) is not increased in the heterozygous mutant cell lines. Furthermore, the telomere blots do not show the heterogenous telomere length patterns typical of ALT lines. The text regarding these issues is as follows:

"The observed telomere elongation is likely due to telomerase-mediated elongation since there was no evidence for increased telomere recombination in the cell lines (Figure 5—figure supplement 3) and the telomeres did not show the typical heterogeneous size of ALT telomeres (Figure 5). An effect of TIN2 on telomerase-mediated telomere elongation is consistent with prior data showing that a truncated TIN2 protein induced dramatic telomere elongation in cells expressing telomerase but had no effect in telomerase-negative cells (Kim et al., 1999)."